

# Simulation of the Ozone Monitoring Instrument Aerosol Index using the NASA Goddard Earth Observing System Aerosol Reanalysis Products

Peter R. Colarco[1], Santiago Gassó[2,3], Changwoo Ahn[1,4], Virginie Buchard[5,6], Arlindo M. da Silva[5], Omar Torres[1]

[1]Atmospheric Chemistry and Dynamics Laboratory, NASA Goddard Space Flight Center, Greenbelt, MD, 20770, USA
[2]Climate and Radiation Laboratory, NASA Goddard Space Flight Center, Greenbelt, MD, 20770, USA
[3]GESTAR/Morgan State University, Baltimore, MD, 21251, USA
[4]Science Systems and Applications Inc., Lanham, MD, 20706, USA
[5]Global Modeling and Assimilation Office, NASA Goddard Space Flight Center, Greenbelt, MD, 20770, USA
[6]GESTAR/Universities Space Research Association, Columbia, MD, 21046, USA

*Correspondence to*: Peter R. Colarco (Peter.R.Colarco@nasa.gov)

**Abstract.** We provide an analysis of the commonly used Ozone Monitoring Instrument (OMI) aerosol index (AI) product for qualitative detection of the presence and loading of absorbing aerosols. In our analysis, simulated top-of-atmosphere (TOA) radiances are produced at the OMI footprints from a model atmosphere and aerosol profile provided by the NASA Goddard Earth Observing System (GEOS-5) Modern-Era Retrospective Analysis for Research and Applications aerosol reanalysis (MERRAero). Having established the credibility of the MERRAero simulation of the OMI AI in a previous paper we describe updates in the approach and aerosol optical property assumptions. The OMI TOA radiances are computed in cloud-free conditions from the MERRAero atmospheric state, and the AI is calculated. The simulated TOA radiances are fed to the OMI aerosol retrieval algorithms, and its retrieved AI (OMAERUV AI) is compared to the MERRAero calculated AI. Two main sources of discrepancy are discussed: one pertaining the OMI algorithm assumptions of the surface pressure, which are generally different from what the actual surface pressure of an observation is, and the other related to simplifying assumptions in the molecular atmosphere radiative transfer used in the OMI algorithms. Surface pressure assumptions lead to systematic biases in the OMAERUV AI, particularly over the oceans. Simplifications in the molecular radiative transfer lead to biases particularly in regions of topography intermediate to surface pressures of 600 hPa and 1013.25 hPa. Generally, the errors in the OMI AI due to these considerations are less than 0.2 in magnitude, though larger errors are possible, particularly over land. We recommend that future versions of the OMI algorithms use surface pressures from readily available atmospheric analyses combined with high-spatial resolution topographic maps and include more surface pressure nodal points in their radiative transfer lookup tables.



## 1 Introduction

The direct radiative effect of atmospheric aerosols changes the energetics of the atmospheric column by scattering and absorption of incident solar and outgoing longwave radiation, generally cooling the underlying surface and possibly warming elevated layers depending on the aerosol absorption properties (e.g., Ångström, 1929; McCormick and Ludwig, 1967;

Chýlek and Coakley, 1974; Charlson et al. 1990, 1991, 1992; Chýlek et al. 1995; Hansen et al. 1997; Haywood et al. 1997). Modification of the temperature profile by this aerosol direct effect has impacts on atmospheric stability and hence clouds (the so-called semi-direct effect, Hansen et al. 1997), and can also feedback on dynamics and so affect the winds and distributions of trace species including water and aerosol and chemical pollutants (e.g., Mulcahy et al. 2014, and see Haywood and Boucher, 2000). The aerosol direct effect depends on the vertical profiles of aerosol loading (usually

represented by the profile of aerosol optical depth, or AOD), aerosol scattering properties (represented by the scattering phase function or more simply by the asymmetry parameter), and the absorption (represented as the single-scattering albedo, or SSA). Obtaining these properties on a global scale is a considerable challenge owing to the spatial, temporal, and compositional (i.e., chemical speciation, size) variability of aerosols. There has been considerable progress in the last 15 years in characterizing the global column-integrated AOD both from ground-based and space-based remote sensing

platforms (e.g., King et al. 1999; Chin et al. 2009). Information on the aerosol vertical profile has also become available in recent years (Welton et al. 2000; Campbell et al. 2003; Winker et al. 2010; McGill et al. 2015), albeit with lesser spatial coverage owing to the active sensor techniques required (i.e., single-beam profiling from ground-based or orbiting lidars). Determination of aerosol phase function is not generally available from remote sensing platforms, although there is some information possible from multi-angle sensors such as the Multi-angle Imaging Spectroradiometer (MISR, Diner et al. 1998)

and the potential for more as multi-angular polarimeters are developed for future missions (e.g., NASA ACE Science Working Group, 2016). Determination of absorption remains, however, a significant challenge, as most satellite remote sensing platforms are only weakly sensitive to this parameter. A recent analysis of estimates of the global direct aerosol radiative forcing highlights aerosol absorption as the largest contributor to overall uncertainty in the direct aerosol radiative effect (Loeb and Su, 2010, see also Kahn, 2011).

One technique for determining column aerosol absorption properties is the near-UV method pioneered in the 1990s with measurements from the space-based Total Ozone Mapping Spectrometer (TOMS, Herman et al. 1997; Torres et al. 1998). The approach determines a qualitative aerosol index (AI) using the observed spectral contrast in two channels where ozone absorption is weak. The AI is a measure of the deviance of the observed spectral contrast from what would be expected in a purely molecular atmosphere. In the absence of clouds, the AI signal has sensitivity to the aerosol loading (i.e., AOD),

altitude, and spectral contrast in single-scattering albedo (Torres et al. 1998, Hsu et al. 1999). The AI approach for detection of absorbing aerosols has been applied to other sensors, including GOME (de Graaf et al. 2005) and SCIAMACHY (de Graaf and Stammes, 2005; Penning de Vries et al. 2009).




The AI is a fundamental intermediate parameter used in the retrieval of aerosol properties derived from measurements taken with the Ozone Monitoring Instrument (OMI, Levelt et al. 2006), as well as its TOMS predecessor (Torres et al. 2002). OMI is the successor to the TOMS series, a joint Dutch-Finnish hyperspectral (270 – 500 nm) imager flying onboard the NASA Aura spacecraft as part of the so-called "A-Train" constellation of polar orbiting satellites, in a sun-synchronous orbit

with a 13:45 (ascending node) local afternoon equator crossing time. OMI has a swath width of 2600 km, obtaining near-daily global coverage, and has a pixel size of 13 x 24 km$^2$ at nadir which extends to about 13 x 150 km$^2$ at the outermost part of the swath. Since mid-June 2007 OMI has suffered a "row anomaly" defect which has degraded its spatial coverage in along-track rows. The impact is minimal in 2007, but worsens in 2008 and appears stable since 2011 (http://projects.knmi.nl/omi/research/product/rowanomaly-background.php), which has overall reduced spatial coverage by

about 50% so that OMI now achieves global coverage every two days.

In this work, we focus on the aerosol index produced in the OMI near-UV aerosol algorithm, called OMAERUV (Torres et al. 2007), which follows from the aerosol index introduced above and is described more completely in Section 2.3. The ultimate objective of our work here is to provide a critical evaluation of the OMAERUV aerosol algorithm using new capabilities for simulating the OMI signals from aerosol model simulations. Colarco et al. (2002) described a simulator for

near-UV aerosols based on the previous TOMS data. Using a chemical transport model with aerosol loading, altitude, and particle size distributions constrained by primarily aircraft and ground-based observations they were able to use their simulator to derive the absorption of Saharan dust aerosols, retrieving the imaginary component of the dust refractive index needed to reproduce the observed AI/AOD ratio. The approach has been significantly improved upon by Buchard et al. (2015, hereafter B15), who generated an entirely new radiance simulator based on the Vector Linearized Discrete Ordinate

Radiative Transfer code (VLIDORT, Spurr 2006) and applied it to simulating the OMI AI from aerosol fields simulated with the NASA Goddard Earth Observing System (GEOS-5) model (Rienecker et al. 2011). Aerosol loading in the GEOS-5 simulation was constrained by assimilation of a AOD derived from the Moderate Resolution Imaging Spectroradiometer (MODIS), as described in B15, the so-called "MERRAero" aerosol reanalysis product (see section 2.1). With a sufficiently constrained atmosphere and aerosol state from the model provided as a "true" state of the natural system, we simulate the

OMI observations and calculate the AI from the MERRAero atmospheric profile. This we will call the MERRAero AI. The MERRAero AI is compared to the OMAERUV AI that is derived using the synthetic MERRAero-produced radiances as the input observables. By comparing the OMAERUV returned AI values to those generated from MERRAero we seek to identify and resolve any discrepancies in the two methods of computing the AI. Since AI is a critical parameter entering the OMAERUV algorithm, this study forms the basis for a subsequent analysis of the OMAERUV AOD and AAOD retrieval

products.



## 2 Methodology

Our approach for evaluating AI follows B15. Using results from the GEOS-5 produced MERRAero aerosol reanalysis (Section 2.1) we simulate the OMI radiances (Section 2.2) and compute a MERRAero-based version of the OMI aerosol index (AI, Section 2.3). The simulated radiances can also be fed into a standalone version of the operational OMAERUV

aerosol retrieval algorithms, from which we obtain a retrieval of aerosol properties based on the synthetic radiances, followed by an extensive analysis of the algorithm performance in the context of an Observing System Simulation Experiment (OSSE). In this paper, we concentrate especially on the calculation of the AI, leaving the retrieval of AOD and AAOD (or, equivalently, SSA) from the synthetic radiances and the OSSE study for a follow-on paper. Unlike B15 where the main goal was to evaluate aerosol absorption properties in MERRAero, here we examine the accuracy of the AI reported

by the OMAERUV product, performing a detailed calculation that uses MERRAero aerosol and meteorological fields as our "nature run." One important simplification we are making is that we assume an entirely cloudless atmosphere. That is, the synthetic radiances at the root of our study include only the impacts of scattering from the surface, the molecular atmosphere, and aerosols.

### 2.1 MERRAero

As described in B15, the MERRAero aerosol reanalysis arises from a "replay" of the GEOS-5 model driven by meteorology from the MERRA atmospheric reanalysis (Rienecker et al. 2011), followed by assimilation of 550 nm AOD retrievals derived from MODIS over ocean and dark target land retrievals. Details of the general AOD data assimilation algorithm can be found in Randles et al. (2017) with MERRAero specifics in B15. In practical terms this means the GEOS-5 model is initialized from the MERRA atmospheric state, performs a six-hour forecast, and subsequently has its meteorology

instantaneously replaced by the balanced atmospheric state of the subsequent MERRA analysis. The MERRA meteorological analyses are inserted into our simulation at 3Z, 9Z, 15Z, and 21Z. The GEOS-5 system includes online aerosols via an implementation of the Goddard Chemistry, Aerosol, Radiation, and Transport (GOCART) module (Colarco et al., 2010), with emissions as described in B15. The GOCART module produces a simulation of the three-dimensional distributions of the mass mixing ratios of dust, sea salt, sulfate, and black and organic carbon aerosols, which serves as a first

guess for the 3-hourly aerosol analysis. With the assignment of lookup-table based optical properties we can translate these mass mixing ratios into optical quantities, such as the spectral AOD, SSA, etc.

The MERRAero-produced atmospheric state and aerosol mass mixing ratio distributions used here are identical to those described in B15. MERRAero was produced for the time period 2002 – 2015 at a global 0.625°x 0.5° longitude by latitude horizontal resolution with 72 vertical levels that vary from terrain following near the surface to pressure following near the

tropopause, with a model top at about 80 km. Based on the analysis in B15 and parallel work we have made two adjustments to the aerosol model lookup-tables for computation of aerosol optical properties. First, we have adjusted the spectral dependence and absorption of dust aerosols as suggested in B15 and introduced dust non-spherical effects as



described in Colarco et al. (2014). For our study this yields dust SSA of about 0.82 at 354 nm and 0.86 at 388 nm. Our second modification is to the optical properties of organic carbon (OC) aerosols in our model. Previously we have used the OPAC refractive (Hess et al. 1998) indices for the OC component, which presumed a weak and spectrally flat absorption for OC in the UV. Based on the analysis in B15 and the recent work by Hammer et al. (2016) we opt in this study to treat our

OC as Brown Carbon (BrC) and reassign its optical properties accordingly. Here, we use the refractive indices from Hammer et al. (2016) for their 100% BrC model (particle density $\rho_p = 1800$ kg m$^{-3}$, final column of their Table 5). This allows for a spectral absorption curve in our simulated radiances that is compatible with the assumptions in the OMAERUV retrieval algorithms. These results produce a SSA for our OC component that is about 0.82 at 354 nm and 0.84 at 388 nm. We stress that this solution is not in all cases ideal. In the implementation of GEOS-5 used in this study we do not

distinguish carbonaceous aerosols from biomass burning sources from those arising from anthropogenic sources, which in general would have different optical properties, so it should be understood that in this study we are assigning the BrC optical properties to simulated OC mass everywhere in our model.

    As in B15 we focus our study on the year 2007, specifically the period June – September 2007, and we remind that the analysis is performed assuming a cloud-free atmosphere.

**2.2 Radiance simulation**

    We simulate the OMI radiances at 354 and 388 nm using VLIDORT from the MERRAero fields following the approach in B15. The OMAERUV algorithms used here are an updated research version of the what was used for the officially released OMAERUV products, and are updated relative to what was used in B15. An important difference in this version versus what was used in B15 is that the spectral surface albedo provided is no longer based on the heritage climatology derived

from the OMI-predecessor TOMS series of instruments (i.e., Herman and Celarier, 1997), but now is determined from a new OMI surface climatology recently developed by the OMI aerosol team that will be implemented in the next release of the OMAERUV products. Additionally, while over land we continue to assume a Lambertian surface, in order to be consistent with corrections made to the over ocean reflectances in the OMAERUV algorithms we treat the ocean surface bidirectional reflectance function according to the Cox-Munk formulation (Cox and Munk, 1954). In principle, ocean surface reflectance

is a function of surface wind speed, which could be provided by our model, but we have here made the simplifying assumption that the wind speed is a constant 6 m s$^{-1}$, a choice consistent with what is used in the OMAERUV retrieval algorithms.

    In summary, MERRAero provides the vertical profiles of aerosol mixing ratio and relative humidity needed to compute the aerosol optical properties, and the vertical pressure profile needed to simulate molecular scattering. The GEOS-5

interface to VLIDORT translates the aerosol and molecular vertical profiles to the profiles of AOD, SSA, and phase scattering matrix input to VLIDORT. All of this information from MERRAero is sampled from the global model at the OMI pixel footprint using the OMI Level 2 product for the same day, which additionally provides needed information on the OMI viewing geometry associated with each pixel (i.e., solar zenith angle, sensor zenith angle, and relative azimuth angle) and the





spectral surface reflectivity. In B15, the OMAERUV quality flags were used to discard cloudy pixels; here we are computing the synthetic radiances at the OMI footprints and providing them to the OMAERUV algorithms as if the atmospheric column is cloud free.

### 2.3 The OMI aerosol index

Fundamental to calculating the aerosol index from the radiances is the computation of the so-called Lambertian Equivalent Reflectivity (LER), which is the surface reflectance that would be needed under a purely molecular atmosphere to explain the actual radiance (observed or simulated) at the top of atmosphere for the actual atmospheric profile. The LER at wavelength λ is computed as in B15 as:

$$LER_\lambda = \frac{I_\lambda - I_\lambda^{Ray}}{T_\lambda^{Ray} + S_\lambda^{Ray}(I_\lambda - I_\lambda^{Ray})},$$    (1)

where $I_\lambda$ is the actual (observed or simulated) radiance, $I_\lambda^{Ray}$ is the calculated molecular-only radiance for assuming a dark surface and the given atmospheric profile, and $T_\lambda^{Ray}$ and $S_\lambda^{Ray}$ are respectively the calculated transmission and spherical albedo for that molecular-only atmosphere. An important distinction between the MERRAero derivation of these parameters and the values used in the OMAERUV algorithms is that the MERRAero parameters are calculated using the modeled pressure profile, whereas in OMAERUV the parameters are interpolated from values pre-computed for a pair of atmospheric

profiles generated assuming surface pressures of 600 hPa and 1013.25 hPa (Torres et al., 2013). The pre-computed OMAERUV lookup tables have dimensions in scattering angle space (resolved into seven solar zenith angle nodes, 14 viewing zenith angle nodes, and 11 azimuth angle nodes) that are interpolated between using Lagrange's method. The two surface pressure nodes are interpolated between linearly in log(pressure) space.

In this version of the OMAERUV algorithm, the LER is corrected by the spectral dependence of the surface as in B15, but

here with two modifications:

$$LER'_{388} = LER_{388} - (Alb_{388} - Alb_{354}) \cdot cf,$$    (2)

where $Alb_{388}$ and $Alb_{354}$ are respectively the surface albedo at 388 and 354 nm. Over land these are just the Lambertian surface reflectivities from the long-term OMI climatology. Over ocean these albedos have here been corrected for the wind speed and viewing geometry-dependent Fresnel reflection of the surface (assuming a fixed surface wind speed of 6 m s$^{-1}$).

Additionally, the factor $cf$ introduces an *ad hoc* correction for spectrally varying background (either cloud or surface) reflectance, needed for conditions of moderate-to-high reflectivity. $cf = 1$ for $LER_{388} < 0.15$, 0 for $LER_{388} > 0.8$, and varies linearly in between; in this experiment, in which we neglect clouds, $LER_{388}$ is almost always $< 0.15$, and so almost always we have $cf = 1$.

Finally, the definition of the Aerosol Index is as in B15:




$$AI = -100 \cdot log_{10} \left( \frac{I_{354}}{I_{354}^{Ray}(LER'_{388})} \right), \qquad (3)$$

where $I_{354}$ is in our case the simulated radiance at 354 nm and $I_{354}^{Ray}$ is the calculated radiance for a molecular scattering atmosphere bounded by a surface reflectance of $LER'_{388}$. Equation 3 assumes that both LERs at 388 and 354 nm are equal.

Again, for OMAERUV the radiance $I_{354}^{Ray}$ is interpolated from pre-computed tables performed for atmospheric profiles

assuming surface pressures of 600 hPa and 1013.25 hPa.

## 3 Given the simulated radiances from MERRAero, do the OMAERUV retrieval algorithms recover the same aerosol index as a first-principle calculation?

For the remainder of this paper we focus on using MERRAero as a representation of a "true" nature state. That is, MERRAero is assumed to provide a sufficiently realistic simulation of aerosol distributions and composition that we can

simulate from those fields the radiances OMI would have observed under its viewing conditions. In the following, we refer to the MERRAero AI as the aerosol index calculated in our OMI simulator using those radiances. Additionally, we propagate those same radiances through the OMAERUV retrieval algorithms, and then recover from the retrieval the aerosol quantities (e.g., AI, AOD, and SSA) that can be compared to the known state MERRAero provided in the first place. Wind et al. (2013; 2016) performed similarly-spirited observation simulation analyses with GEOS-5 based on the MODIS aerosol

and cloud algorithms. We focus in this paper only on the AI. In the following, we refer to the OMAERUV AI as the aerosol index returned by the OMAERUV algorithms using the MERRAero computed radiances inputs. Our objective is to identify and then attempt to resolve major areas of discrepancy between the MERRAero AI and OMAERUV AI. To do this we simulate MERRAero radiances for the full OMI swath assuming clear-sky conditions. We do this for all OMI orbits for the months June – September 2007.

### 3.1 Impact of surface pressure

From Equations 1 and 3 we see that the AI depends on a calculation of the atmospheric molecular scattering, which in turn depends on an assumption of the atmospheric pressure profile. The assumption of the atmospheric pressure profile is handled differently in OMAERUV versus in MERRAero. In OMAERUV, the surface pressure is nominally assumed to have a fixed value of 1013.25 hPa at sea level that is adjusted by a high-spatial resolution topography dataset in order to

account for differences in the molecular scattering over high mountains versus the oceans, and the molecular atmosphere radiance is recovered by interpolating the pre-computed atmospheric radiance lookup tables between their 600 hPa and 1013.25 hPa values, as discussed in Section 2.3. In MERRAero, the atmospheric pressure profile used is based on the model grid-box mean surface pressure and the radiative transfer of the molecular atmosphere is solved exactly for that pressure profile. It is important to point out that with the approximately 50 km grid-scale of MERRAero we are certainly not





resolving the actual spatial variability in topography that is assumed in the OMAERUV pressure scaling. For our purposes, it does not matter that MERRAero does not resolve those high-spatial resolution topographic features since our objective is simply to put MERRAero and OMAERUV on as much of the same footing as possible, but we will revisit this point later.

Figure 1a shows the MERRAero AI on June 5, 2007, showing major absorption features across Saharan Africa, the
Arabian Peninsula, and much of southern Asia. Other absorption features are present over southern Africa, in the Pacific Ocean immediately west of Mexico, and near Beijing and across the northern Pacific. Note the wide areas, mainly over the ocean, that are shaded grey. Because the OMAERUV algorithms rely on pre-computed lookup tables of the atmospheric radiance profile bounded by surface pressures of 600 and 1013.25 hPa we have here and for the remainder of this paper excluded grid points where the MERRAero grid-box mean surface pressure is outside of those bounds. Figure 1c shows the
OMAERUV – MERRAero difference in the AI, both using the MERRAero simulated radiances. The differences in the AI are small, mainly less than 0.2 in magnitude, and not obviously associated with the aerosol features shown in Figure 1a. Mostly the AI differences are positive (OMAERUV AI > MERRAero AI, green shading), although there are some regions showing negative AI differences (MERRAero AI > OMAERUV AI, red shading) over the land in southern Australia and northern Europe.

For Figure 1c, the OMAERUV AI is computed with the algorithm using its default assumption of the surface pressure. Thus the assumed surface pressure in the OMAERUV algorithm is constant over the oceans and static in time, and so differs from the actual surface pressure at any given moment, where spatial and temporal variability in the surface pressure in the MERRAero simulation is assured by the MERRA meteorology (i.e., surface pressure in the real atmosphere changes with weather patterns). Figure 1b shows this OMAERUV – MERRAero difference in the surface pressures on June 5, 2007, and
it is clear that differences of tens of hPa are possible. Negative pressure differences (red shading) in Figure 1b are apparent over the land, mainly in southern Australia and northern Europe, and so coincident with the negative AI differences shown in Figure 1c. (It is worth noting here that most of the greyed out region over the oceans occurs in places where MERRA has surface pressures greater than 1013.25 hPa, and so are screened out in this analysis. See supplemental Figure S1 for a version of Figure 1 where the surface pressure screening is not applied).

In Figure 1d we show the OMAERUV AI – MERRAero AI difference where the OMAERUV calculation was performed using the MERRAero grid-box mean surface pressure, and indeed we see that much of the AI difference apparent in Figure 1c is reduced in magnitude in Figure 1d. The negative bias over the land is much reduced, as is the noise near topographic features. Residual positive discrepancies in the AI as high as $\Delta AI \sim 2$ remain, however, and these seem mainly to be associated with regions of high topographic variability. For the approximately 538,000 AI pixels on this day, only about 5%
of them have an AI discrepancy greater than 0.2 for the calculation where the OMAERUV AI is computed using the MERRAero pressure, versus 20% for the case where OMAERUV used its own pressure.

Figure 2 presents a similar analysis for the entire month of June 2007. In Figure 2a we present the joint histogram of the OMAERUV AI – MERRAero AI difference for the case of OMAERUV calculating AI with its own surface pressure assumption versus the OMAERUV – MERRAero surface pressure difference for all OMI pixels during the month (again,



excluding those where the MERRA surface pressure is out of bounds of the OMAERUV lookup tables). It is clear in Figure 2a that most of the pixels are near the origin, where the differences on both axes are nearly zero. The excursions toward high surface pressure differences are relatively infrequent (10s to 100s of points, versus the 100,000s of points with near-zero pressure differences) and are mainly associated with the difference of the MERRAero resolved surface pressure from the

OMAERUV values in highly variable topography (i.e., the noise in the mountainous regions shown in Figure 1b). Also clear from Figure 2a is the sense in which a difference in the assumed surface pressure propagates to a difference in the derived AI. Where OMAERUV assumes a higher surface pressure than MERRAero it also systematically derives a higher AI. The opposite case is also true: OMAERUV derives a lower AI for pixels in which its surface pressure is assumed lower than MERRAero provides.

Figure 2b presents the same results sorted differently, showing the AI difference as a function of the MERRAero AI. Colored points are for the case where the OMAERUV AI is calculated using the OMAERUV surface pressure, with the color indicating the OMAERUV – MERRAero surface pressure difference. Grey points are for the case where OMAERUV AI is calculated using the MERRAero surface pressures. It is immediately apparent that much of the scatter in the AI difference is reduced when OMAERUV calculates the AI using the MERRAero surface pressure. Most of the discrepancy

in the OMAERUV AI calculation where OMAERUV uses its own surface pressure occurs for low positive values of the MERRAero AI, with low AI values indicating either low aerosol loading, low aerosol altitude, or else the presence of non-absorbing aerosols. The discrepancy is also clearly a function of the surface pressure discrepancy, with the greatest negative AI differences occurring at pixels where the surface pressure difference is most negative, and vice versa for the high AI differences. On the other hand, at high values of the MERRAero AI the AI difference is nearly zero. It follows that as

aerosol loading decreases the radiative transfer becomes more sensitive to the molecular profile, and so especially to differences in the pressure profile implied by the OMAERUV surface pressure relative to what is provided by MERRAero. For the baseline case of OMAERUV using its own surface pressure, 18% of the pixels have an absolute difference in the simulated AI from MERRAero greater than 0.2. When OMAERUV uses the MERRAero surface pressure only 5% of the pixels have an absolute difference in the AI greater than 0.2. The results for the full month shown in Figure 2 are thus very

similar to the results for the single day showed in Figure 1.

Figure 3 continues this analysis, showing maps of the monthly mean OMAERUV AI – MERRAero AI difference for June 2007 for both the cases where OMAERUV calculates AI using its own surface pressure (Figure 3a) and where OMAERUV uses the MERRAero grid-box mean surface pressure (Figure 3b). Over the ocean there are large regions that are again excluded (grey shading) because MERRAero persistently has surface pressure > 1013.25 hPa (Figure S2 shows the same

figure but without the high pressure areas screened). The AI differences are generally positive (OMAERUV AI > MERRAero AI) and can be as large as 0.1 − 0.2 over the oceans, and greater than 0.2 over the land. The only apparent negative differences are in southern Australia again. (For comparison, Figure S2 highlights the dominant high pressure regions over the ocean in the MERRA meteorology; although the OMAERUV retrieval of AI when forced with these high surface pressures is extrapolated outside the bounds of the lookup tables the sense is that these are places where substantial



negative AI biases would exist in the actual OMAERUV products.) In Figure 3b, where the MERRAero surface pressure was used to calculate the OMAERUV AI, the over-ocean AI discrepancy is greatly reduced. Over land the residual AI difference is larger and mostly positive (i.e., OMAERUV > MERRAero). Figure 3b shows also the 1000 m height contour over the land surface, from which it is apparent that the residual AI difference is spatially correlated with topographic features. The results are similar for our analysis of July, August, and September (Figure 4, see also Figure S3), with the land residual difference remaining basically the same from month to month, while the ocean residual moves with the changing months.

### 3.2 Impact of radiative transfer interpolation

Section 3.1 identified the surface pressure assumptions as a significant driver of the MERRAero AI and OMAERUV AI differences given the same input radiances. When OMAERUV was forced to use the MERRA-provided surface pressure the residual AI differences were greatly reduced, as shown in Figures 3 and 4, particularly over the ocean. As shown in Figures 3b and 4, however, the AI residual over land in cases where OMAERUV used the MERRAero surface pressure remains and is a persistent, stationary feature, apparently spatially correlated with topographic features. We explore here the nature of this residual difference. Our hypothesis is that this difference results largely from differences in the treatment of the molecular atmosphere scattering between MERRAero and OMAERUV, as discussed above in Section 2.3.

We examine this here for a region in the central United States, extending from roughly western Illinois to western Nevada (longitude from 120° – 90° W, latitude from 38° – 42° N) and transitioning from relatively flat, lower elevations in the east to higher elevations crossing the Rocky Mountains. Figure 5a shows the mean and one-standard deviation about the mean of all the MERRAero and OMAERUV coincident aerosol index points in this region during June 2007, sorted by longitude. Again, here the OMAERUV AI is computed using MERRAero surface pressure. The high bias of the OMAERUV aerosol index relative to MERRAero is consistent with Figures 3b and 4, and shows this bias increasing toward the west; i.e., for the higher topography portion of the region. This is made explicit in Figure 5b where we show the OMAERUV AI – MERRAero AI difference plotted with the topographic elevation. The correlation of the AI difference with elevation is clear.

Having identified a high bias in the OMAERUV AI over land with respect to MERRAero and finding an apparent correlation with surface elevation, we explore further with a simple sensitivity analysis. We construct a synthetic orbit of the OMI Level 2 data in which a spectrally invariant surface albedo of 0.05 (a typical value for the surface reflectivity) is prescribed and we define a range of viewing geometries that encompasses typical OMI viewing angles (solar zenith angle between 8° – 96°; sensor zenith angle between 0° and 70°; relative azimuth angle of 0°, 90°, and 180°). The atmosphere is assumed aerosol free; that is, we only consider molecular scattering. We perform calculations of the top-of-atmosphere (TOA) radiance with the MERRAero OMI simulator code for each view geometry considered and for variations in 50 hPa increments of the surface pressure between 1000 hPa and 600 hPa. As in the previous AI comparisons we provide the simulated TOA radiances to the OMAERUV algorithms and look at the derived parameters returned. The MERRAero



calculations return AI = 0 and LER = 0.05. This result is expected by construction of the problem: in a molecular-only atmosphere with a spectrally invariant Lambertian surface the LER is equivalent to the surface albedo and the AI is expected to be zero by Equation 3.

Figure 6 presents the results of this sensitivity analysis as applied to the OMAERUV calculation. The LER as derived in

Equation 1 depends on the "observed" TOA radiance (i.e., what MERRAero provides) and properties of the molecular atmosphere: $I_\lambda^{Ray}$, $T_\lambda^{Ray}$, and $S_\lambda^{Ray}$. Differences between the MERRAero and OMAERUV transmissivity are very small, but more sizable differences in the spherical albedo and atmospheric radiance are found. Figures 6a and 6b show these as a percent difference (OMAERUV – MERRAero)/MERRAero for the spherical albedo and atmospheric radiance $I_{388}^{Ray}$, respectively, presented as a function of scattering angle and surface pressure. Differences are very small at the end-point

surface pressures at 600 hPa and 1000 hPa, but are about 1.5% in the spherical albedo and 2.5% in the atmospheric radiance at the mid-point 800 hPa surface pressure. These errors percolate through the calculation so that the LER from OMAERUV is not identical to the surface albedo (= 0.05) as expected, but instead is too low, varying between about 0.04 and 0.05, as shown in Figure 6c. Finally, the bias in the AI is revealed in Figure 6d, where the error in the LER propagates through to a high bias in the OMAERUV AI that can be as much as 0.5 at scattering angles near 120° and surface pressures near 700 hPa.

More typically the AI bias is smaller, but the sensitivity analysis performed offers an explanation of the AI bias shown over the central US (and more generally over land) in Figure 5. We contend that ultimately this error emerges from the simple interpolation of the pre-computed atmospheric radiance calculations between the 600 hPa and 1013.25 hPa nodal points used in the OMAERUV algorithms. We note additionally that there is an apparent scattering angle sensitivity to the results shown in Figure 6. This applies particularly to the atmospheric radiance (Figure 6b) and the resultant aerosol index (Figure 6d).

Because the differences shown in Figure 6 are based on the interpolation from the two pressure nodes of the OMAERUV lookup tables it follows that the apparent scattering angle sensitivity also results from the interpolation of the lookup tables to the specific viewing geometries used in our analysis. We do not explore this further here, but suggest that additional nodes in the OMAERUV lookup tables will mitigate these kinds of uncertainties.

## 4 Conclusions

We have updated and expanded on the capabilities of the OMI radiance simulator described in B15. The OMI TOA radiances at 354 and 388 nm were calculated from the same MERRAero aerosol profiles as used in B15. The AI formulation was updated from B15, and we introduced the viewing geometry dependent Fresnel reflectance at the ocean surface and its correction as implemented in the current OMAERUV algorithms. In order to improve the realism of the MERRAero simulated radiances we have updated the optical property assumptions of dust and organic aerosols as described in Section

30   2.1.

The OMI TOA radiances were simulated for the period June – September 2007, and the MERRAero AI was computed from the model results. The radiances were provided to the OMAERUV aerosol retrieval algorithms, which returned its own





calculation of the AI, which was subsequently compared to those derived from MERRAero. Two major discrepancies were identified:

1) The assumed surface pressure in the current OMAERUV algorithms results in a systematic error in the OMAERUV retrieved AI to the extent that that surface pressure differs from the actual surface pressure. This was shown in Figures 1 – 4 by comparing calculations where OMAERUV used its default surface pressure assumptions with calculations used the surface pressure provided by MERRAero. Where the OMAERUV surface pressure was greater than the MERRAero surface pressure, the default OMAERUV calculation resulted in a higher AI than MERRAero, most notably over land. Where MERRAero had a higher surface pressure than OMAERUV assumed the result was the opposite. Mostly these differences are small, with less than about 18% of the pixels will have errors exceeding 0.2 in AI magnitude, but there is a clear association of these errors with the presence of weather systems, particularly over the ocean. We do not here assess how this plays out in the actual OMI AI products where of course the presence of clouds in the real scenes will mask the impact of this discrepancy. Nevertheless, when OMAERUV is forced to retrieve AI using the time varying surface pressures provided by MERRAero we find a significant improvement in the AI difference, reducing to only at 5% the number of pixels with residual AI differences greater than 0.2 in magnitude.

2) Following the analysis of the surface pressure discussed above we found that the residual differences remaining were associated with topographic features over land and were similar from month to month. We note that the OMAERUV AI relies on pre-computed lookup tables of atmospheric radiances fields, including transmissivity and spherical albedo of the molecular-only atmosphere. These lookup tables are provided to the OMAERUV algorithms valid at surface pressures of 600 hPa and 1013.25 hPa. Intermediate pressures are derived by linear interpolation from these nodal points in log(pressure) space to the topographic height for the selected pixel. A sensitivity analysis (Figure 6) shows differences in the molecular atmosphere calculated fields between MERRAero and OMAERUV that are generally small near the nodal-point pressures and largest at intermediate pressures. Propagating these errors through to the calculation of AI we find that AI errors can be as large as 0.5 depending on the surface pressure and viewing geometry.

These results allow us to make two recommendations with respect to the OMAERUV algorithms. First, the surface pressure assumptions used in the operational algorithms should be revisited and consideration should be given to using readily available surface pressures from meteorological analyses from any modern weather prediction system (i.e., from the GEOS-5 near-real time prediction system). A caveat to that recommendation is that the surface pressure from the analysis model must be modulated by a high-spatial resolution tropography dataset in order to provide surface pressures consistent with the actual viewing conditions over variable terrain. This was not done in our study, where we used the grid-box mean surface pressure values. The second recommendation is that the molecular atmosphere calculation should be revisited to either incorporate a more realistic, exact radiative transfer solution with respect to the actual surface pressure and atmospheric profile, or else to at least include more intermediate nodal points in the lookup table solutions to improve





accuracy in the returned AI. Both of these recommendations are under consideration for future versions of the OMAERUV aerosol products.

The analysis presented in this paper has demonstrated the use of a state-of-the-art aerosol modeling system to simulate radiances observed by real observing systems. We used those simulated fields to interrogate the aerosol retrieval algorithms

applied to the data from those observing systems. It should be noted that the magnitude of the AI discrepancies found in this study are typically small, generally less than 0.1 over the ocean, but can be larger (often 0.5 or higher) over land. While these discrepancies may not be large in terms of the semi-quantitative way in which the AI is often used in the research community, we point out that the AI is used to threshold certain algorithmic choices in the OMAERUV retrievals of AOD and AAOD, and so we expect these discrepancies to have non-negligible impact on those products. A subsequent study will

follow-up our approach further and critically examine the retrieved AOD and AAOD from the OMAERUV algorithms. Further development of these capabilities will facilitate development of new observing systems and algorithm development by revealing important sensitivities of algorithm and observation choices.

## Acknowledgements

This work was carried out under the auspices of a NASA Aura ROSES proposal, NNH13ZDA001N-AURA, Ken Jucks

(program manager). Calculations of the MERRAero simulated atmospheric radiances were performed at the NASA Center for Climate Simulation (NCCS) computing cluster. The intermediate version of the OMAERUV research algorithms used in this study are version 1.6.2.1.

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




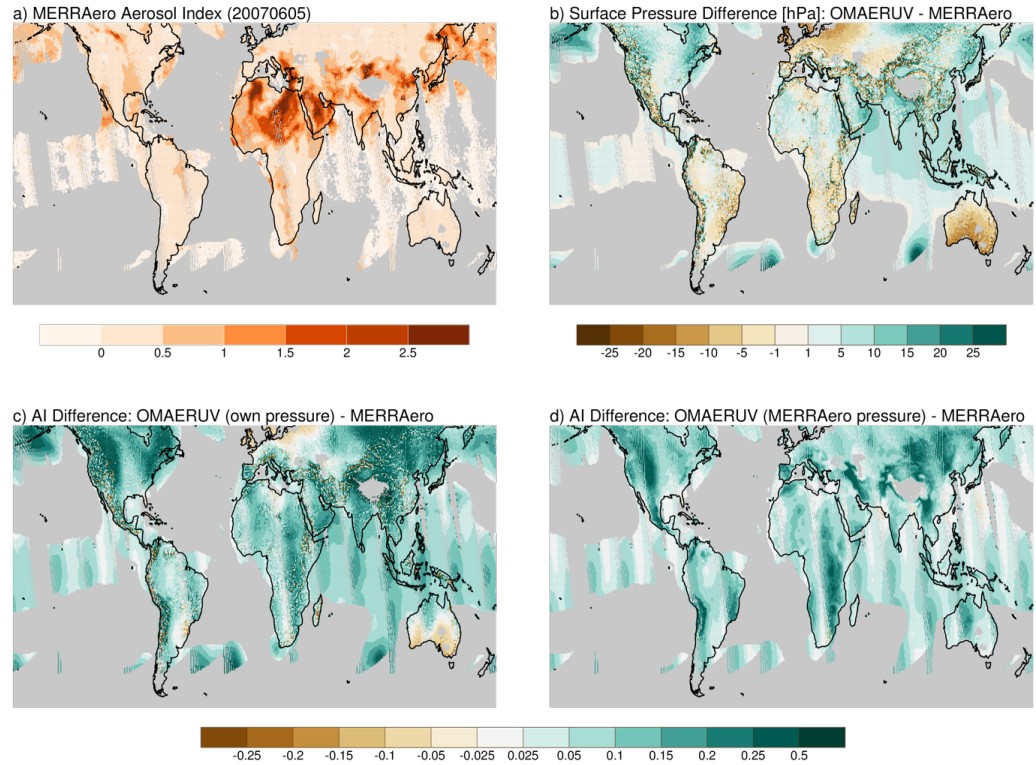

**Figure 1: (a) MERRAero simulated aerosol index on June 5, 2007. (b) OMAERUV – MERRAero difference in surface pressure [hPa] for the same day. (c) OMAERUV AI – MERRAero AI difference, with OMAERUV using its own surface pressure. (d) OMAERUV AI – MERRAero AI difference with OMAERUV using the MERRAero provided surface pressure.**





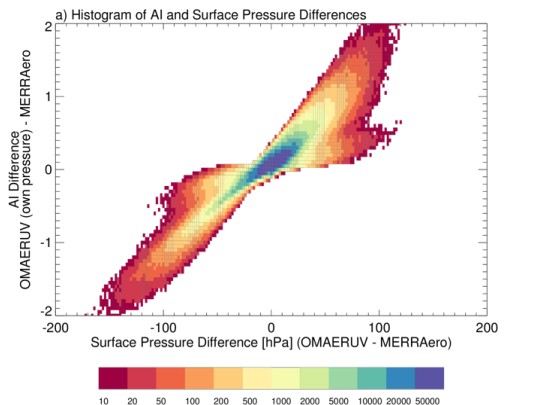
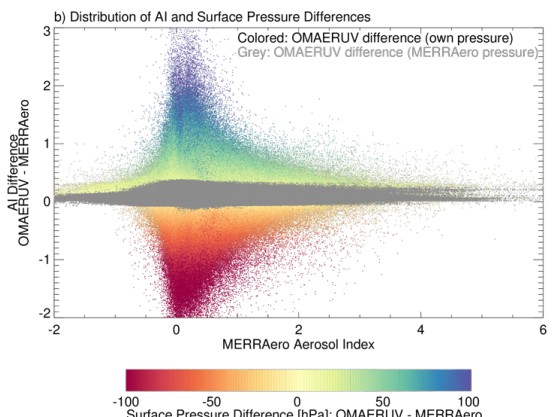

**Figure 2: (a) Frequency distribution of OMAERUV AI – MERRAero AI differences versus OMAERUV – MERRAero surface pressure differences for all pixels in June 2007. (b) OMAERUV AI – MERRAero AI difference as function of the MERRAero AI for all pixels during June 2007. Colored points are for AI difference when OMAERUV uses its own surface pressure, with the coloring indicating the OMAERUV – MERRAero surface pressure difference. Grey points are for the AI difference residual when OMAERUV calculates AI using the MERRAero surface pressure.**




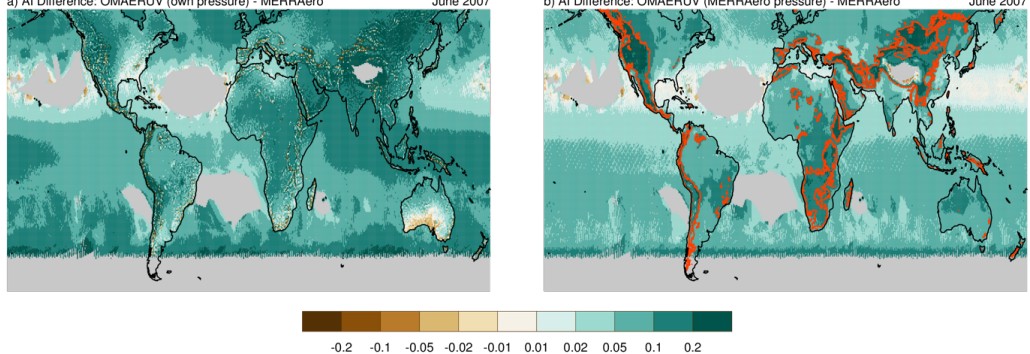

**Figure 3: OMAERUV – MERRAero AI differences for June 2007. (a) OMAERUV AI calculated using its own surface pressure. (b) OMAERUV AI calculated using MERRAero surface pressure. The red line is the 1000 m topographic height contour.**





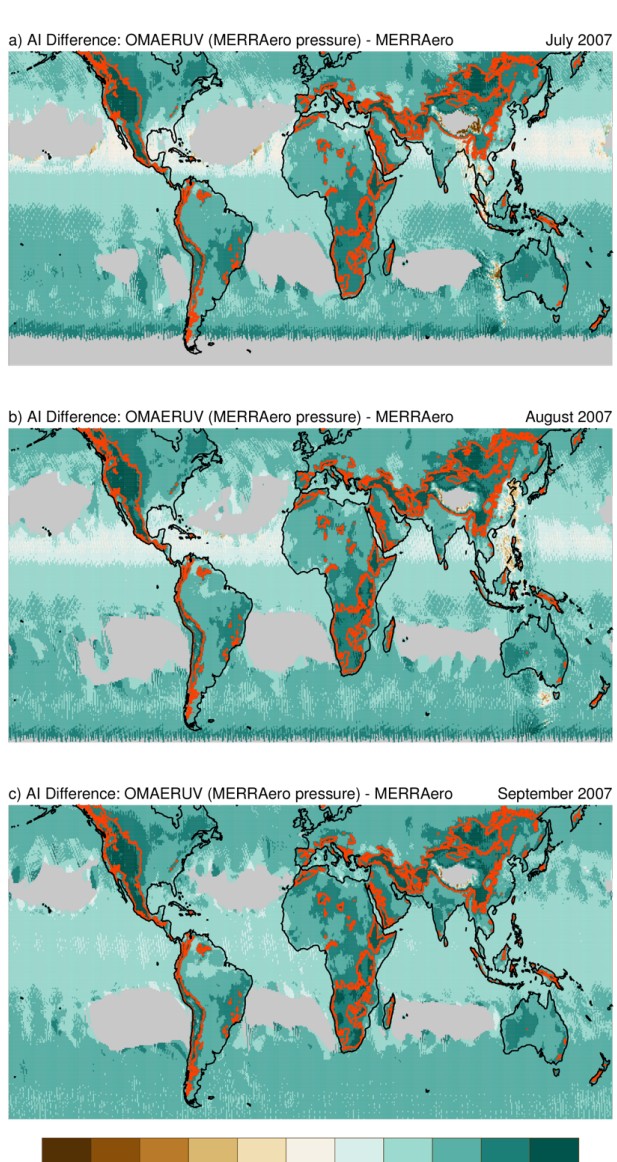

**Figure 4: As in Figure 3b, but showing the OMAERUV – MERRAero AI difference for OMAERUV using the MERRAero surface pressure for (a) July, (b) August, and (c) September 2007. Again, the white line is the 1000 m topographic height contour.**




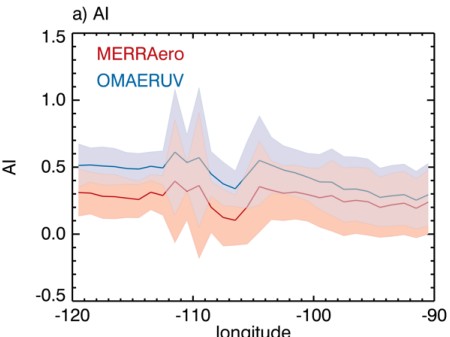

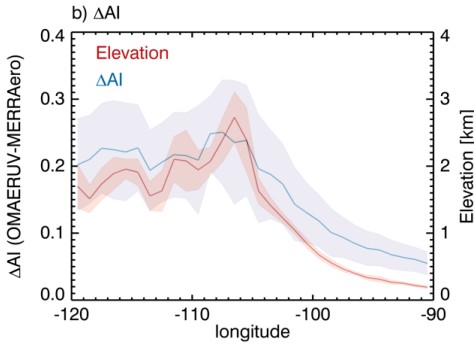

**Figure 5: (a) MERRAero and OMAERUV aerosol index mean and one standard deviation range for all retrievals in the box enclosing longitude from 120° – 90° W, latitude from 38° – 42° N, over the central United States during June 2007. In all, 36855 data points were considered. The OMAERUV AI shown here is computed using the MERRAero surface pressure. (b)**
5  **OMAERUV – MERRAero AI residual mean difference and one standard deviation about the mean over the same region (blue) and the regionally averaged topographic elevation and one standard deviation about the mean (red).**



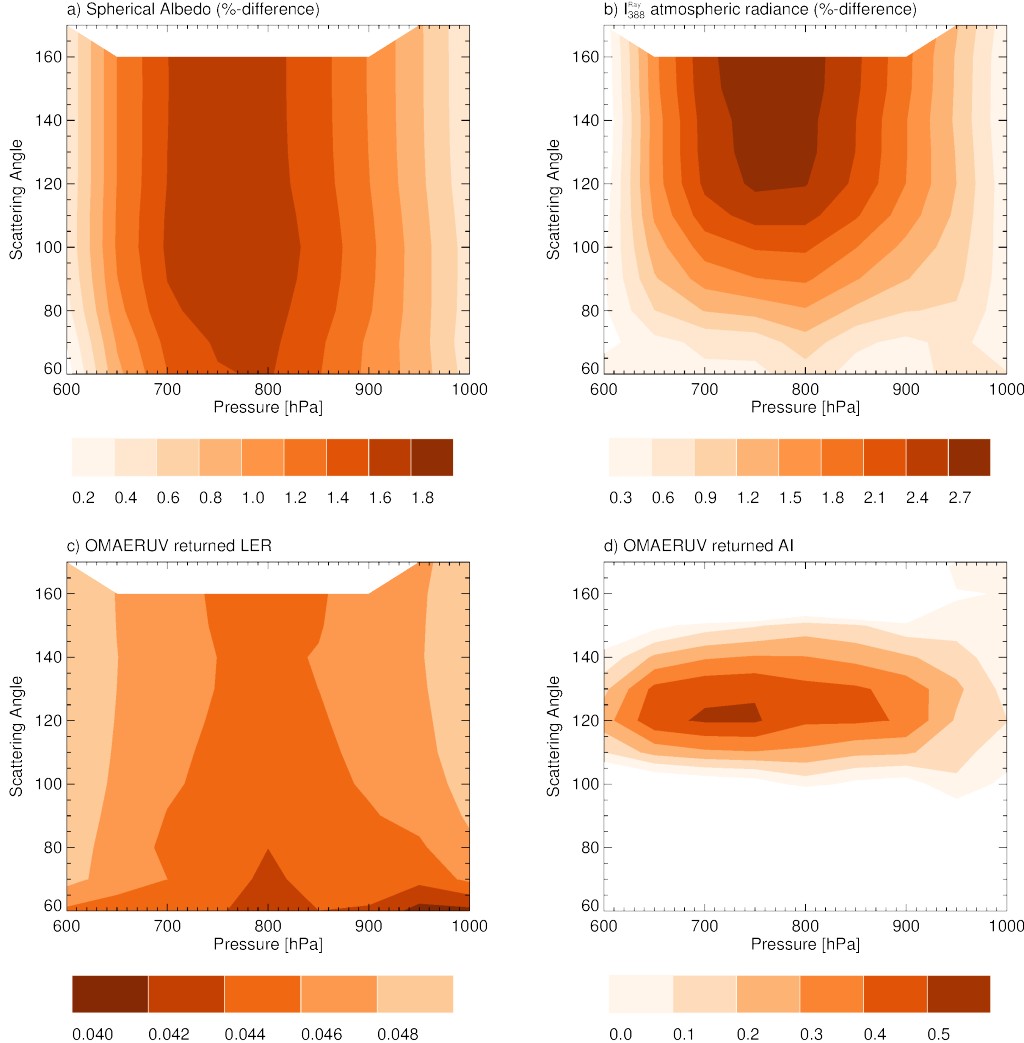

**Figure 6: Results of sensitivity study exploring pressure interpolation of radiative transfer calculation results provided by OMAERUV. (a) Percent difference in spherical albedo, calculated as (OMAERUV-MERRAero)/MERRAero. (b) Percent difference in molecular atmosphere radiance at 388 nm, calculated as (OMAERUV-MERRAero)/MERRAero. (c) LER derived from OMAERUV calculations. (d) Aerosol index derived from OMAERUV calculations.**