# Peer review of "Simulation of the Ozone Monitoring Instrument Aerosol Index using the NASA Goddard Earth Observing System Aerosol Reanalysis Products"

_Atmospheric Measurement Techniques, 2017_

## Referee Comment (RC1) · Anonymous Referee #1 · 7 Jun 2017

This paper uses MERRAero simulations of OMI radiances to generate an Aerosol Index (AI) based on OMAERUV algorithm assumptions (OMAERUV AI) and compare the results to an AI generated with appropriate (MERRAero generated) values of surface pressure and molecular radiative transfer. In light of prior literature by the first author and others, this work can be considered an incremental improvement. That said, the paper presents a justification for practical improvements in the OMAERUV algorithm, and in that sense is important. It is also very well written and organized. I recommend it for publication with minor technical modifications.

Specific comments:

[Figure]

If I understand correctly, this analysis assumes no OMI measurement uncertainty on the part of either the MERRAero AI or the OMERAUV AI. I imagine there must be some characterization of the OMI random errors (via SNR) and systematic biases (via calibration tracking). Since you're working with large datasets, you could 'add' such measurement errors into your simulated observations. The reason for doing this would be to put the OMERAUV algorithm related biases and errors in context. I could imagine a scenario where it would make sense not to add the complexity required for the suggested changes because measurement uncertainty generates a larger product error. Of course, I would hope this is not the case, but it would have been nice to see this.

I appreciate seeing the modifications that were made to the OPAC properties to account for dust and brown carbon. That said, can you discuss what the implications of non-realistic aerosol properties in this analysis? Is it important just to span the range of possible aerosol conditions and to do so in a way that mimics the frequency of occurrence in nature? Could the conclusions change if you get this wrong?

One conclusion of this study is that OMERAUV could incorporate surface pressure fields from a weather prediction system. If you're doing that, you might as well also take surface winds to drive the glint over the ocean, rather than use the constant 6m/s. To some extent I was expecting to see an analysis of the consequences of this assumption.

Was there any specific reason for the choice of June-September timeframe for the analysis? Could any difference be expected for other seasons of the year?

There's a season dependent residual over the ocean, which means a geometric dependent bias. This is also indicated by the apparent swath dependent biases. Figure 6 indicates biases that are expressed in a geometrically dependent manner. Would fixing the lookup tables in the radiative transfer make all the geometrically depended biases go away?

The map figures show between 60ËŽ north and south. Are OMI retrievals not performed above 60ËŽ?

Typos: Page 9, line 24: "showed" -> shown Figure 4, caption: "white line" -> "red line"

———————————————

---

## Referee Comment (RC2) · M.J.M. Penning de Vries (Referee) · 21 Jul 2017

The authors describe the investigation of systematic errors in the current version of the OMAERUV AI product. Several years' worth of OMI radiances were simulated from realistic model aerosol scenarios and were fed both into the OMAERUV algorithm, and processed into a "true" AI for comparison. Good agreement was found for both products, but systematic differences on the order of 0.2 units were also observed. The main conclusion from the paper is that the quality of the current version of the algorithm is affected by a lack of nodes in surface pressure and angle (solar zenith, viewing, and relative azimuth angle) space.

Although the technical approach and the used methods are valid and the study appears to be soundly performed, one is left with the question if the conclusions could not have been found in a much simpler way, involving much less computing time. Investigating if look-up-table (LUT) interpolation is sufficiently accurate is a rather trivial exercise, for which several tens of radiative transfer calculations should suffice. Can the authors comment on this, and possibly add a few lines to the manuscript explaining why such an extensive study was set up?

As the paper is technically and scientifically sound, I recommend it for publication if the comment above and the minor suggestions below are sufficiently addressed.

Page 2, line 10: pedantically, AOD is the integrated extinction. Hence "AOD profile" is inaccurate, but should read "extinction profile"

Page 2, lines 15-24: Please mention: that MISR measures aerosol height of optically thick layers; that ESA is planning the 3MI instrument, which is also dedicated to aerosol properties; that many aerosol characteristics have been obtained from the POLDER instrument (e.g. by Dubovik's GRASP algorithm, but also earlier by Waquet and co-workers)

Page 2, line 30: The AI is sensitive to the absolute values and spectral dependences of both AOD and SSA, in addition to altitude

Page 2, line 32: And OMPS on Suomi-NPP

Page 5, line 17: "updated research version of the what was used" - please correct

Page 5, Section 2.2.: please comment on sun-glint. I assume this is not simulated - or is it?

Page 6, Section 2.3.: What method is used for LUT interpolation?

Page 6, line 25: What does cf stand for? I was confused with cloud fraction, but this would have the opposite effect. You might consider using another symbol.

Page 7, Section 3.1: Was the Tibetan Plateau also screened? It appears grey in all figures.

Page 12, conclusion 1: If the dependence of the AI difference on pressure is linear, can we use the old OMAERUV results and simply correct them using this observed dependence?

---

## Author Comment (AC1) · 1 Sep 2017

This paper uses MERRAero simulations of OMI radiances to generate an Aerosol Index (AI) based on OMAERUV algorithm assumptions (OMAERUV AI) and compare the results to an AI generated with appropriate (MERRAero generated) values of surface pressure and molecular radiative transfer. In light of prior literature by the first author and others, this work can be considered an incremental improvement. That said, the paper presents a justification for practical improvements in the OMAERUV algorithm, and in that sense is important. It is also very well written and organized. I recommend it for publication with minor technical modifications.

**We thank the reviewer for the time and consideration of our paper, as well as the constructive comments.**

Specific comments:

If I understand correctly, this analysis assumes no OMI measurement uncertainty on the part of either the MERRAero AI or the OMERAUV AI. I imagine there must be some characterization of the OMI random errors (via SNR) and systematic biases (via calibration tracking). Since you're working with large datasets, you could 'add' such measurement errors into your simulated observations. The reason for doing this would be to put the OMERAUV algorithm related biases and errors in context. I could imagine a scenario where it would make sense not to add the complexity required for the suggested changes because measurement uncertainty generates a larger product error. Of course, I would hope this is not the case, but it would have been nice to see this.

**The reviewer is correct: we did not characterize the impact of any random errors in the simulated OMI radiance signals. While this would be interesting, the objective of the paper was focused on the impact of algorithmic choices particularly in the OMAERUV codes on the retrieved aerosol quantities. We have added the following text to Section 3 (page 7, lines 27 – 30):**

**"We do not assume any errors in the simulated radiances provided as input to the OMAERUV retrieval algorithms. While inclusion of random, realistic errors in the simulated radiances would further the characterization of the OMAERUV algorithmic performance the focus of this paper was rather on the algorithmic choices and theirs impacts on the retrieved aerosol quantities."**

I appreciate seeing the modifications that were made to the OPAC properties to account for dust and brown carbon. That said, can you discuss what the implications of non-realistic aerosol properties in this analysis? Is it important just to span the range of possible aerosol conditions and to do so in a way that mimics the frequency of occurrence in nature? Could the conclusions change if you get this wrong?

**In Section 2.1 we discuss the adjustment made here to the OPAC properties was based on the work in Buchard et al. 2015 (B15, Buchard, V., A. M. da Silva, P. R. Colarco, A. Darmenov, C. A. Randles, R. Govindaraju, O. Torres, J. Campbell, and R. Spurr (2015), Using the OMI aerosol index and absorption aerosol optical depth to evaluate the NASA MERRA Aerosol Reanalysis, *Atmos Chem Phys*, 15(10), 5743–5760, doi:10.5194/acp-15-5743-2015).  As seen in B15 (for dust: their Figures 7 & 8 and Table 2; for smoke: their Figures 12 & 13 and Table 4) the choice of magnitude and spectral dependence in the imaginary component of the refractive index affects the simulation of the AI, as well as other derived quantities such as absorbing aerosol optical depth (AAOD) and aerosol direct radiative forcing.  It is the relatively better agreement found between the simulated and actual OMI retrieved quantities in B15 that justifies the particular refractive index choices made in our study.**

**We did not explore the sensitivity of our results to different assumptions of the aerosol optical properties than those we made here.  As seen in B15 if we used more OPAC-like properties we would have had different AAOD for dust and the attendant impact on the radiance in dusty regions, for example, or a different spectral dependence in the radiance in the biomass burning regions.  These would have affected the results in some way, but probably not the main points we investigated that were related to surface pressure and lookup table resolution as they relate instead to, respectively, the molecular scattering with respect to the aerosol layer and the details of the radiative transfer.  On the other hand, a possible impact presents itself in that the retrieval of AAOD and AOD (which is not considered in this study) does depend on prescribed models in the OMAERUV algorithms.  If our simulations are using optical properties that are not in the space the optical properties assumed in OMAERUV then possibly we would find large differences in the retrieved and simulated AAOD and AOD that would be the result of this discrepancy rather than other algorithmic choices.  We have thought about this, and the assumed spectral dependences and magnitudes of the imaginary refractive index used in this study for the various aerosol components in the model are compatible with the range of aerosol models used in the current OMAERUV algorithms.  The specific impact of these choices will have to be evaluated in a subsequent study.  We have added the following text to Section 2.1 to address this question (page 5, lines 15-23):**

**"The implications of these choices should be modest here because of the limited nature of our study.  The model fields are being used to simulate the top-of-atmosphere radiance field, and since we limit this study to the investigation of the OMI aerosol index our particular choices of refractive indices should be sufficiently realistic for that purpose.  On the other hand, in the case that we extend this analysis to a critical evaluation of the OMAERUV AAOD and AOD products it will be important that our choices of refractive index for the forward**

model simulation of the radiances is compatible with the aerosol models assumed in the OMAERUV retrievals (Torres et al. 2007).  In anticipation of that future study this is indeed the case, but do not investigate this further here."

One conclusion of this study is that OMERAUV could incorporate surface pressure fields from a weather prediction system. If you're doing that, you might as well also take surface winds to drive the glint over the ocean, rather than use the constant 6m/s. To some extent I was expecting to see an analysis of the consequences of this assumption.

**Relaxing the assumption of the surface wind speed over the ocean proved to be less straightforward than we originally thought.  While we could have computed the simulated TOA radiances using the actual surface wind speeds, and then followed through on the analysis, this would not have itself resolved the central question here.  The issue is that surface LER over ocean is corrected in the OMAERUV retrievals for Fresnel reflectance as if the wind speed is 6 m s$^{-1}$ based on a pre-computed correction table.  In order to resolve the question what impact this assumption has on the AI it would have put the burden on algorithm team to generate this correction for arbitrary actual surface wind speeds, which was unfortunately beyond the scope of the work we could carry out here.  So we opted for the simple approach used here.  We clarify this in the paper in Section 2.2 (page 5, beginning line 38):**

**"In principle, ocean surface reflectance is a function of surface wind speed, which could be provided by our model, but we have here made the simplifying assumption that the wind speed is a constant 6 m s$^{-1}$, a choice consistent with what is used in the OMAERUV retrieval algorithms, where the observed radiances are corrected for surface reflectance based on the climatological OMI surface reflectance with an imposed Fresnel correction as if the surface wind speed was 6 m s$^{-1}$ (see Section 2.3)."**

**and in Section 2.3 (page 6, beginning line 34):**

**"Over ocean these albedos have here been corrected for the wind speed and viewing geometry-dependent Fresnel reflection of the surface based on a pre-computed table assuming a fixed surface wind speed of 6 m s$^{-1}$.  We did not investigate this assumption as the requirement to produce a Fresnel correction lookup table at arbitrary wind speed was beyond the scope of this work, but it should be stressed that the correction applied in the OMAERUV algorithm is consistent with what is done in the MERRAero AI calculation given the same radiances."**

Was there any specific reason for the choice of June-September timeframe for the analysis? Could any difference be expected for other seasons of the year?

**The June – September timeframe was chosen because it picked up main features of Saharan dust transport and African and South American biomass burning and was consistent with the**

**analysis in B15. We add to the conclusions (page 13, lines 14 – 19):**

**"It should be noted that our analysis was performed for a single season (June – September 2007) under simulated aerosol loadings expected to be valid in that season. Differences in the aerosol loading, composition, and vertical distribution at other times of the year may have some effect on the conclusions presented here, although we expect the main points to hold. A possible seasonal dependence in the over-ocean residual AI difference following the surface pressure correction was found in Figure 4 and should be explored further."**

There's a season dependent residual over the ocean, which means a geometric dependent bias. This is also indicated by the apparent swath dependent biases. Figure 6 indicates biases that are expressed in a geometrically dependent manner. Would fixing the lookup tables in the radiative transfer make all the geometrically depended biases go away?

**Possibly, although the scattering angle dependency in the AI (Figure 6d) probably has more relationship to any scan angle biases than seasonal biases since the full range of scattering angles can be found at essentially any latitude. We have added the following text to the end of Section 3.1 (page 10, lines 19 – 21):**

**"We do not explore the geometric nature of this small seasonal dependence further here. Possibly it would be resolved by improved lookup tables of the OMAERUV radiative transfer (see next Section) or could be characterized further by simulating a longer period of time (e.g., an annual cycle)."**

The map figures show between 60 north and south. Are OMI retrievals not per formed above 60?

**The retrievals are performed at all sunlit latitudes. We have modified the figures to show this.**

Typos: Page 9, line 24: "showed" -> shown Figure 4, caption: "white line" -> "red line"

**Corrected, thank you.**

Reviewer #2: M.J.M. Penning de Vries (Referee)

The authors describe the investigation of systematic errors in the current version of the OMAERUV AI product. Several years' worth of OMI radiances were simulated from realistic model aerosol scenarios and were fed both into the OMAERUV algorithm, and processed into a "true" AI for comparison. Good agreement was found for both products, but systematic differences on the order of 0.2 units were also observed. The main conclusion from the paper is that the quality of the current version of the algorithm is affected by a lack of nodes in surface pressure and angle (solar zenith, viewing, and relative azimuth angle) space.

Although the technical approach and the used methods are valid and the study appears to be soundly performed, one is left with the question if the conclusions could not have been found in a much simpler way, involving much less computing time. Investigating if look-up-table (LUT) interpolation is sufficiently accurate is a rather trivial exercise, for which several tens of radiative transfer calculations should suffice. Can the authors comment on this, and possibly add a few lines to the manuscript explaining why such an extensive study was set up?

**We thank the reviewer for the time and attention to our paper, as well as the constructive comments. We agree that some sensitivities of the OMAERUV algorithms could indeed have been explored with relatively simple radiative transfer calculations. On the other hand, what our approach affords is an opportunity to systematically and comprehensively explore various aspects of the retrieval algorithm, of which the aerosol index is only the first instance. In the future, we will apply our approach to investigate the AOD and AAOD provided by the OMAERUV retrievals. Having the known truth from a model-provided "nature" state is fundamental to the exercise, as the model provides a reasonably realistic variability in aerosol composition and temporal and spatial (both horizontal and vertical) distributions which will enhance the analysis and evaluation. Finally, demonstrating this capability enhances the confidence in applying the model as a tool in developing and assessing proposed future satellite observation concepts. (We note that, in contrast to what the reviewer has written, we did not draw any conclusion about the angular resolution in the lookup tables; rather, the sensitivity analysis in Figure 6—shown in terms of scattering angle—reflects more on the angular sensitivity of the pressure interpolation).**

**We have modified the text in Section 1 to emphasize the application (page 3, beginning line 39):**

**"Since AI is a critical parameter entering the OMAERUV algorithm, this study forms the basis for a subsequent analysis of the OMAERUV AOD and AAOD retrieval products to be performed in the future. Additionally, the methodology used here has the more general application of laying out an approach for using a well-constrained, realistic chemical transport model as a known "nature" state to simulate the observations of future satellite instruments and observing systems."**

As the paper is technically and scientifically sound, I recommend it for publication if the comment above and the minor suggestions below are sufficiently addressed.

Page 2, line 10: pedantically, AOD is the integrated extinction. Hence "AOD profile" is inaccurate, but should read "extinction profile"

**Corrected, thank you.**

Page 2, lines 15-24: Please mention: that MISR measures aerosol height of optically thick layers; that ESA is planning the 3MI instrument, which is also dedicated to aerosol properties; that many aerosol characteristics have been obtained from the POLDER instrument (e.g. by Dubovik's GRASP algorithm, but also earlier by Waquet and co- workers)

**We have modified the text accordingly (page 2, lines 21 – 33):**

**"Determination of aerosol phase function is not generally available from remote sensing platforms, although there is some information possible from multi-angle sensors such as the Multi-angle Imaging Spectroradiometer (MISR, Diner et al. 1998) and the potential for more as multi-angular polarimeters are developed for future missions (e.g., NASA ACE Science Working Group, 2016, and the European Space Agency 3MI instrument manifested for launch on METOP SG-A in mid-2021). The MISR instrument additionally provides estimates of aerosol height for optically thick layers by exploiting its stereo viewing capabilities (e.g., Mims et al. 2010). Determination of absorption remains, however, a significant challenge, as most satellite remote sensing platforms are only weakly sensitive to this parameter, although work done with the recent space-based Polarization and Directionality of Earth Reflectances (POLDER, Waquet et al. 2016) and Polarization and Anisotropy of Reflectances for Atmospheric Sciences coupled with Observations from a Lidar (PARASOL, Lacagnina et al. 2015) instruments, and polarimetry gives some insight into this, as it does also for aerosols above clouds (Peers et al. 2015)."**

Page 2, line 30: The AI is sensitive to the absolute values and spectral dependences of both AOD and SSA, in addition to altitude

**We have modified the text (beginning page 3, lines 1 – 3):**

**"In the absence of clouds, the AI signal has sensitivity to the aerosol loading (i.e., AOD, including its spectral dependence), altitude, and spectral contrast in single-scattering albedo (Torres et al. 1998, Hsu et al. 1999)."**

Page 2, line 32: And OMPS on Suomi-NPP

**We have added a reference to that.**

Page 5, line 17: "updated research version of the what was used" - please correct

**This is correct as written.**

Page 5, Section 2.2.: please comment on sun-glint. I assume this is not simulated - or is it?

**The OMAERUV algorithms screen for sunglint regions over the ocean and do not retrieve AI (or other aerosol properties) where the glint angle is < 20°. Accordingly we screen the simulated MERRAero AI results using the same criteria.**

Page 6, Section 2.3.: What method is used for LUT interpolation?

**Here is what we wrote (page 6, lines 22 – 25):**

**"The pre-computed OMAERUV lookup tables have dimensions in scattering angle space (resolved into seven solar zenith angle nodes, 14 viewing zenith angle nodes, and 11 azimuth angle nodes) that are interpolated between using Lagrange's method. The two surface pressure nodes are interpolated between linearly in log(pressure) space."**

Page 6, line 25: What does cf stand for? I was confused with cloud fraction, but this would have the opposite effect. You might consider using another symbol.

**"cf" is a "correction factor" which could also be related to cloud fraction, applied as described in the text to weight the contribution of the spectral surface albedo difference to the LER. The factor is largest (greatest contribution of spectral surface albedo difference) in cases where the surface signal is large (i.e., lower reflectivity) and goes to zero in cases where the surface contribution to the signal is reduced (i.e., higher reflectivity, typically cases where clouds would dominate the scene). We have renamed the factor "f" at the reviewer's suggestion.**

Page 7, Section 3.1: Was the Tibetan Plateau also screened? It appears grey in all figures.

**The Tibetan Plateau is screened in our analysis because the surface pressure atop the mountain peaks is lower than the 600 hPa minimum in the OMAERUV radiative transfer lookup tables. We added the following text (page 7, lines 20 – 21):**

**"Note the wide areas, mainly over the ocean, but also including the Tibetan plateau, that are shaded grey."**

Page 12, conclusion 1: If the dependence of the AI difference on pressure is linear, can we use the old OMAERUV results and simply correct them using this observed dependence?

**We would be cautious in drawing that conclusion because as Figure 2a shows although the relationship between the surface pressure and AI differences is highly linear it is not perfectly so, and neither would we expect it to be since the atmospheric signal is also related to the**

vertical profile of the aerosol with respect to the molecular background.  So a proper correction would involve a reprocessing of the AI products with analysis-provided surface pressures.